# Differential Diagnosis of Thoracoacromial Artery Pseudoaneurysm from Shoulder Inflammatory Pseudotumor: A Case Report

**DOI:** 10.3390/diagnostics13010082

**Published:** 2022-12-28

**Authors:** Tzu-Yen Huang, Pin-Chao Feng, Yao-Chang Wang, Chun-Yi Su

**Affiliations:** 1Department of Thoracic and Cardiovascular Surgery, Chang Gung Memorial Hospital at Keelung, Keelung 20401, Taiwan; 2Department of Biomedical Engineering, National Taiwan University, Taipei 10617, Taiwan; 3Department of Thoracic and Cardiovascular Surgery, Chang Gung Memorial Hospital at Linkou, Taoyuan City 33302, Taiwan; 4Department of Orthopaedic Surgery, Chang Gung Memorial Hospital, Keelung Branch, Bone and Joint Research Center, Keelung 20401, Taiwan; 5Department of Orthopaedic Surgery, Chang Gung University, Taoyuan City 33302, Taiwan

**Keywords:** shoulder tumor, pseudoaneurysm, inflammatory pseudotumor, angiogram, endovascular coiling

## Abstract

Diagnosing shoulder tumors is a challenge because the joint is very complex, and a static examination can misdiagnose some tumors. However, we found that a pseudoaneurysm provides a differential diagnosis of a tumor, and to that end, we present a case that mimics shoulder infection. The patient was an 80-year-old female who had a history of coronary artery disease and end-stage renal disease under regular hemodialysis and complained of right shoulder swelling and progression. A magnetic resonance imaging (MRI) scan revealed an abscess, inflammatory pseudotumor (IPT), and osteomyelitis of the humerus. Computed tomography (CT)-guided pigtail drainage was performed twice without significant improvement. An angiogram revealed a right shoulder pseudoaneurysm fed by the acromial branch of the thoracoacromial artery. After endovascular coiling, the patient was discharged and outpatient follow-up was arranged. If aspiration of the abscess leads to only mild improvement in shoulder swelling, further evaluation should be arranged. An angiogram examination is a good method for diagnosing and designing operations, and endovascular treatment is good for preventing injury to the muscle, tendon, nerve, or blood vessels.

## 1. Introduction

The structure of the shoulder is very complex because it is the body’s most mobile joint [1,2]. Most shoulder tumors can be diagnosed by X-ray or magnetic resonance imaging (MRI), but these are static evaluations, which is the main problem with diagnosing shoulder diseases [3] because they could lead to misdiagnoses of, for example, cancer, tracheomalacia, or a pseudoaneurysm [3,4,5,6,7,8].

The tumors of the shoulder should be diagnosed carefully. Fifteen percent of primary bone sarcomas are diagnosed in the shoulder [9]. The presentations of shoulder lesions usually result from one of four reasons: pain, a lump, an incidentally found lesion, or a pathologic fracture [10]. Imaging, laboratory evaluation, and biopsy can help diagnose shoulder tumors and help us with staging if the tumor is a malignant tumor [10].

The wall of an artery has three layers: intima, media, and adventitia. A pseudoaneurysm occurs when the artery has lost its adventitia and peri-adventitial tissue [11]. Most shoulder pseudoaneurysms have traumatic or iatrogenic causes [12,13,14,15,16,17,18], but an infection, such as an abscess, leads to the destruction of the adventitia layer, which eventually causes a pseudoaneurysm [11].

Pseudoaneurysms of the shoulder are rare, and their diagnosis is challenging [7,12]. An arterial pseudoaneurysm usually results from penetrating trauma [13,14], shoulder dislocation, or fracture [12,15,16,17]. It could also have an iatrogenic cause, such as arterial catheterization [12], arthroscopy [7,18], and arthroplasty [19]. In rare cases, infections from vessel injuries also cause pseudoaneurysms [20,21].

A person with an aneurysm may be asymptomatic, but when symptoms do emerge, it is usually because of the aneurysm’s mass effects on peripheral nerves, organs, or vessels. Pseudoaneurysms progress in size and lead to rupture and massive bleeding [22,23]. Correct diagnosis and treatment to prevent irreversible neurological injury or life-threatening bleeding are very important.

This report discusses a rare case of a shoulder pseudoaneurysm caused by abscess on a rare lesion, which was hardly noticed due to no trauma or surgical history. The differential diagnosis of this case was difficult but finally had good outcomes. We acquired information using various approaches, including X-ray, laboratory evaluation, MRI, and several procedures, thus, the whole process was long. Therefore, we would like to share our experience to help accelerate similar diagnoses in the future.

## 2. Case Presentation

An 80-year-old female with a history of coronary artery disease, triple-vessel disease, and end-stage renal disease under regular hemodialysis complained of right shoulder swelling that kept progressing. Passive exercises exacerbated the pain and range of motion was limited. There was severe local tenderness and local heat, but the patient declared no history of trauma or surgery. An X-ray suggested no orthopedic damage, and a hematology test revealed no leukocytosis or left shift.

The orthopedist then followed up with an MRI scan on the first day after admission and noticed a 3.3 × 8.2 cm abscess in front of the humeral head extending across subacromial and subdeltoid space with intra-articular involvement. The scan also showed a 1.7 cm inflammatory pseudotumor (IPT) within the supraspinatus muscle, and osteomyelitis of the humerus (Figure 1). The antibiotics vancomycin and ceftriaxone were prescribed upon admission. On the sixth and thirty-second days after the MRI scan, computed tomography (CT)-guided pigtail drainage was arranged, which only mildly decreased the pain and swelling (Figure 2). Because the improvement to the shoulder mass and the symptoms was rather slow, a surgical shoulder debridement procedure was arranged to remove blood clots and insert antibiotic beads. During the operation, a pulsatile tumor was noticed. The orthopedist did not complete the tumor resection or tumor biopsy for the risk of tumor rupture combining uncontrolled bleeding. Under the impression it was a pseudoaneurysm, the orthopedist team decided to complete the debridement without touching it.

The cardiovascular department was consulted for further treatment, and an angiogram of the right shoulder was arranged. It revealed a pseudoaneurysm fed by the acromial branch of the thoracoacromial artery (Figure 3). An operation to embolize the pseudoaneurysm proceeded with the patient in a supine position with her right arm stretched out. A minimal incision was made in the brachial artery above the elbow and a 10 cm, 6 Fr. endovascular therapy sheath (TERUMO CORPORATION, Radifocus Introducer II M, Tokyo, Japan) was inserted. Endovascular surgery was performed on the thoracoacromial feeder artery with a 0.89 mm (in diameter) × 220 cm (in length) Terumo guidewire (TERUMO CORPORATION, Radifocus guidewire M, Tokyo, Japan) and a 100 cm JB1 sheath (Cook Medical, JB1 sheath, Bloomington, IN, USA). Three embolization tornado coils (Cook Medical, Tornado Embolization Coil, Bloomington, IN, USA) were used to embolize the feeding vessel, and at the end of the surgery, an intra-operation angiogram showed complete occlusion with no contrast enhancement in the pseudoaneurysm (Figure 4).

After embolization, the patient reported that the pain in her shoulder was much improved, and that there was no more progression of the swelling (Figure 5). She was discharged on the twenty-fifth day after embolization and followed up with the outpatient department.

## 3. Discussion

Most pseudoaneurysms are caused by penetration injuries, even if they are the result of a needle puncture injury during nerve block [22,24]. Shoulder or axillary open procedures and arthroscopy would cause iatrogenic pseudoaneurysm [18,24,25,26,27]. In this case, the thoracoacromial pseudoaneurysm was caused by local inflammation with abscess. Pseudoaneurysms are seldom caused by local inflammation [22]. For such cases, there could be a high risk of misdiagnosis when assessed using image examinations due to the aneurysm being very close to the inflammation and cause artifacts. For our case, it was initially noted as an IPT [28,29].

In previous studies, pain and swelling were the most noted symptoms of shoulder pseudoaneurysms [7,18,19,24]. The symptoms of the pseudoaneurysm in this case were also pain and swelling on the lesion. There was no obvious mass effect, such as nerve compressions or left arm numbness and palsy. After the shoulder abscess was treated twice by CT-guided pigtail drainage and once by shoulder debridement, the patient still had not recovered. We decided to investigate the IPT, which was also caught on the same MRI image. An IPT could lead to a nonspecific diagnosis, but its features could help prevent unnecessary procedures [28,29]. A biopsy would be helpful for a differential diagnosis, but it is invasive with a high risk of bleeding and infection. A dynamic image examination may provide more diagnostic information.

Since an X-ray or MRI offers only a static evaluation [3] of a pseudoaneurysm or any other shoulder disease, a dynamic examination (sonogram [3], angiogram, dynamic CT, or dynamic MRI) [4,5,7] offers a more comprehensive perspective, especially concerning joints. In our case, considering the IPT on the MRI image and the pulsatile tumor found in the debridement, we used an angiogram to explore the possibility of a pseudoaneurysm. As the abscess was the root cause of the pseudoaneurysm, it was located next to it. Artefacts of the abscess may have affected the contours of the pseudoaneurysm, so the image could be estimated as an IPT. A sonogram could give us abundant dynamic information, but in this case, the lesion was in a deep area that is difficult for a sonogram to access. Dynamic CT, dynamic MRI, computed tomography angiography, magnetic resonance angiography, and angiogram can also help diagnose a shoulder tumor. We chose an angiogram because it could find the specific feeding vessel, decrease the amount of contrast medium, and design our operation. In the end, we noticed that the pseudoaneurysm was in the same place as the IPT on the MRI image. In our case, we arranged an early dynamic examination after the shoulder abscess drainage resulted in poor healing.

There are two ways to treat pseudoaneurysm: surgical excision to remove the pseudoaneurysm and endovascular embolization to block the blood supply to the pseudoaneurysm [22,23]. Surgical excision requires the separation of muscles, tendons, nerves, and vessels [22]. In endovascular embolization, instead of directly approaching the shoulder, a guidewire and a sheath are inserted into the brachial artery for coiling or injection of embolization agents. Because of the complex structure of the shoulder, we chose endovascular embolization, which has become increasingly popular because the endovascular devices and implants are well-established [7,19,22,23,24].

## 4. Conclusions

Differential diagnosis of shoulder tumors is a very important issue due to the possibility of malignant tumors [10]. Most of the lesions bring pain or lumps. The initial evaluation of shoulder tumors mainly decides the approaches used in treatment, which could vary from simply medication for inflammation to surgical procedures for pseudoaneurysms or cancers. Pseudoaneurysms are not malignant tumors. However, a ruptured pseudoaneurysm could be a lethal situation [22].

Because of the shoulder’s complexity, a pseudoaneurysm should serve as a differential diagnosis for a tumor. It can be caused by infection as well as trauma or iatrogenic vessel injury [20,21]. The IPT noted on the MRI was well established, which ruled out a dynamic and lethal disease. When the abscess aspiration resulted in only mildly improved shoulder swelling, an angiogram was considered to locate the source and assess the extent of the problem. We would also suggest other types of dynamic examination to be considered in other cases.

Surgery is the only treatment for a shoulder pseudoaneurysm, and an endovascular embolization is a good method for preventing injury to the muscle, tendon, nerve, or vessel, and offers good long-term follow up.

## Figures and Tables

**Figure 1 diagnostics-13-00082-f001:**
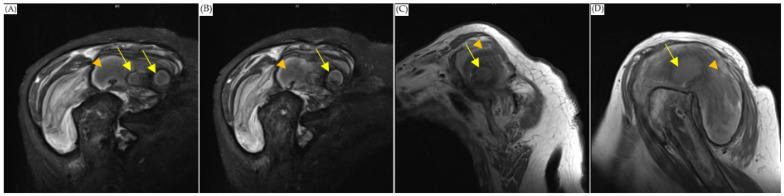
MRI of shoulder. (**A**,**B**) were coronary plane, (**C**,**D**) were sagittal plane. All the images showed: 1. An encapsulated cystic mass extending across subacromial and subdeltoid space (arrowhead), which was suggestive of abscess. 2. Enhancing nodule within the supraspinatus muscle, which was suggestive of inflammatory pseudotumor (arrow).

**Figure 2 diagnostics-13-00082-f002:**
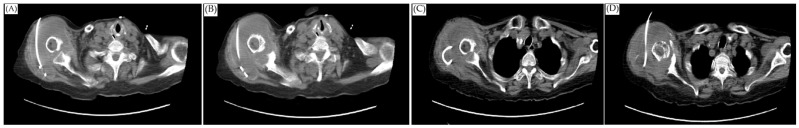
CT-guided drainage of the abscess. (**A**,**B**) were the first CT-guided pigtail drainage; (**C**,**D**) were the second CT-guided pigtail drainage; the abscess of the subdeltoid space was mildly decreased. However, local swelling only mildly decreased.

**Figure 3 diagnostics-13-00082-f003:**
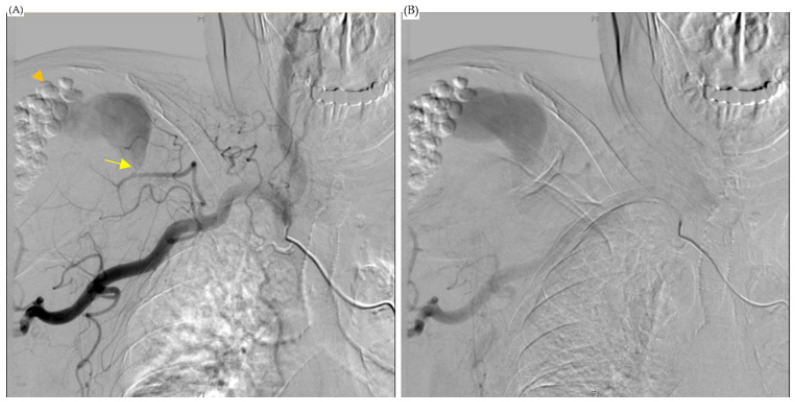
Angiogram showed a pseudoaneurysm. (**A**) Angiogram showed that the feeding artery of the pseudoaneurysm was the thoracoacromial artery, acromial branch (arrow). Antibiotic beads were noted in this angiogram (arrowhead). (**B**) The aneurysm extended to acromion of scapula.

**Figure 4 diagnostics-13-00082-f004:**
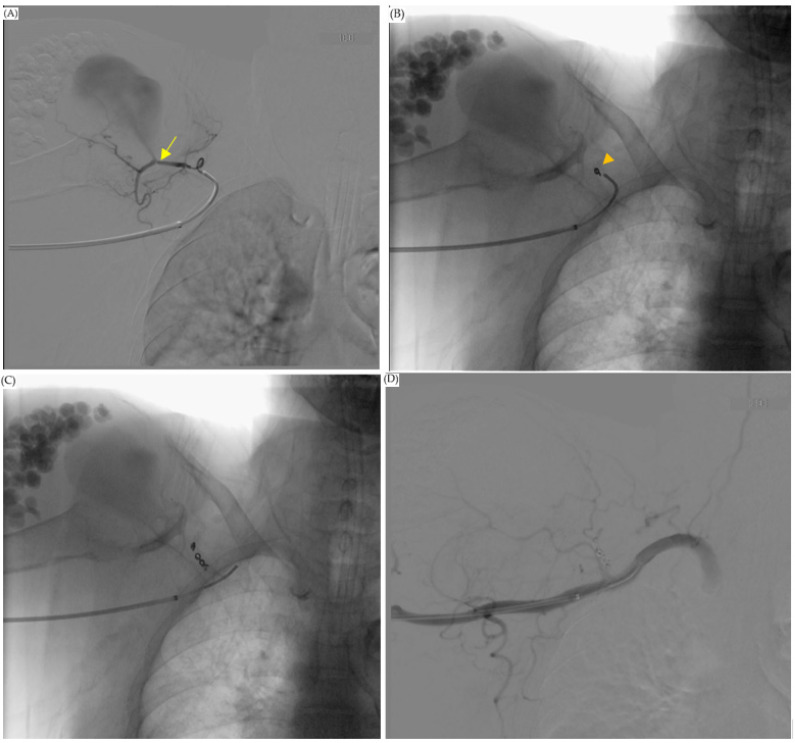
Operation: embolization of the thoracoacromial artery, acromial branch. (**A**) Guidewire with JB1 catheter attached to the acromial branch. Contrast medium injected from JB1 catheter showed we entered our target. The aneurysm stump was also detected (arrow). (**B**) We used Tornado coils (arrowhead) to coil the feeding artery. (**C**) Three Tornado coils were inserted in the thoracoacromial artery, acromial branch. (**D**) At the end of the operation, angiogram showed total embolization of the feeding artery. There was no contrast pooling into the pseudoaneurysm.

**Figure 5 diagnostics-13-00082-f005:**
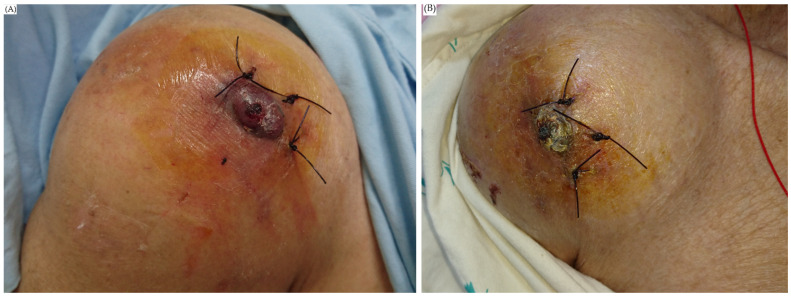
Right shoulder of the patient. (**A**) On the day before endovascular embolization, local heat, swelling and redness of the right shoulder were noted. The stiches were left from debridement. (**B**) One week after endovascular embolization, her shoulder was much improved in terms of swelling and pain.

## Data Availability

The data presented in this study are available on request from the corresponding author.

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
