# Peer review of "Differential Diagnosis of Thoracoacromial Artery Pseudoaneurysm from Shoulder Inflammatory Pseudotumor: A Case Report"

_diagnostics, 2022, doi:10.3390/diagnostics13010082_

Round 1

Reviewer 1 Report

The case report is interesting, mostly well written.

Minor considerations:

Line 32 – what about the histopathology in tumor diagnostics?

Conclusions should be improved.

Institutional Review Board Statement: This study was approved by the institutional review board 167 of the ethical committee of Chang Gung Memorial Hospital. – a number should be provided

Author Response

Response to Reviewer 1 Comments

Point 1:  Line 32 – what about the histopathology in tumor diagnostics?

Response 1: Thank you. This patient didn’t undergo tumor resection or biopsy to avoid the risk of bleeding. I added line 80 and 81 in the case presentation to explain the reason.

Point 2: Conclusions should be improved.

Response 2: Thank you. I have added line 178 to 183 in the conclusion to further explain why the differential diagnosis of such cases are important and what risks and consequences could be if not done carefully.

Point 3: Institutional Review Board Statement: This study was approved by the institutional review board 167 of the ethical committee of Chang Gung Memorial Hospital. – a number should be provided

Response 3: Thank you. Pleas find the IRB number 202201646B0. I have added this in line 200.

Thank you very much for the advice and reminders. This not only has helped me improve this report but also helped me in academic writing in the furture.

Reviewer 2 Report

This was a case report focused on a thoracoacromial artery pseudoaneurysm in a patient suffering from shoulder inflammatory pseudotumor. I think the article should be improved in some respects:

- In the introduction section you should add the purpose of the study or at least specify the scientific contribution of this article to the literature

- the first paragraph of the discussion seems to be more suitable for introduction

- the discussion section is a bit poor in arguments. You should try to increase the themes that emerged from this case report, also making comparisons with the articles already present in the literature.

Author Response

Response to Reviewer 2 Comments

Point 1:  In the introduction section you should add the purpose of the study or at least specify the scientific contribution of this article to the literature

Response 1: Thank you. I have added line 36 to 40 and line 56 to 62 in the introduction to address the three main points the case could contribute as to how this puedoaneurysm was diffifult to diagnose, which are the location (shoulder), the root cause (local inflammation), and the examination approaches (the limitation of static imaging).

Point 2: the first paragraph of the discussion seems to be more suitable for introduction

Response 2: Thank you. I have moved this paragraph to line 41 to 45 in the introduction.

Point 3: the discussion section is a bit poor in arguments. You should try to increase the themes that emerged from this case report, also making comparisons with the articles already present in the literature.

Response 3: Thank you. I have searched more literatures for the common causes of pseudoaneurysms and others’ experiences. However, there are limited reports on shoulder aneurysms. I have added line 136 to 143 to conclude my finding from these literatures.

Thank you very much for the advice and reminders. This not only has helped me improve this report but also helped me in academic writing in the furture.